# Intraduodenal Delivery of Exosome-Loaded SARS-CoV-2 RBD mRNA Induces a Neutralizing Antibody Response in Mice

**DOI:** 10.3390/vaccines11030673

**Published:** 2023-03-16

**Authors:** Quan Zhang, Miao Wang, Chunle Han, Zhijun Wen, Xiaozhu Meng, Dongli Qi, Na Wang, Huanqing Du, Jianhong Wang, Lu Lu, Xiaohu Ge

**Affiliations:** 1Tingo Exosomes Technology Co., Ltd., Tianjin 300301, China; zhangquan@tingocell.com (Q.Z.); wangmiao@tingocell.com (M.W.); hanchunle@tingocell.com (C.H.); wenzhijun@tingocell.com (Z.W.); mengxiaozhu@tingocell.com (X.M.); qidongli@tingocell.com (D.Q.); wangna@tingocell.com (N.W.); duhuanqing@tingocell.com (H.D.); wangjianhong@tingocell.com (J.W.); 2Tingo Regenerative Medicine Technology Co., Ltd., Tianjin 300301, China

**Keywords:** bovine-milk-derived exosomes, SARS-CoV-2, receptor-binding domain, mRNA, oral vaccines, neutralizing antibodies

## Abstract

Severe acute respiratory syndrome coronavirus type 2 (SARS-CoV-2), which causes coronavirus disease 2019 (COVID-19), has presented numerous challenges to global health. Vaccines, including lipid—based nanoparticle mRNA, inactivated virus, and recombined protein, have been used to prevent SARS-CoV-2 infections in clinics and have been immensely helpful in controlling the pandemic. Here, we present and assess an oral mRNA vaccine based on bovine-milk-derived exosomes (milk-exos), which encodes the SARS-CoV-2 receptor-binding domain (RBD) as an immunogen. The results indicate that RBD mRNA delivered by milk-derived exosomes can produce secreted RBD peptides in 293 cells in vitro and stimulates neutralizing antibodies against RBD in mice. These results indicate that SARS-CoV-2 RBD mRNA vaccine loading with bovine-milk-derived exosomes is an easy, cheap, and novel way to introduce immunity against SARS-CoV-2 in vivo. Additionally, it also can work as a new oral delivery system for mRNA.

## 1. Introduction

COVID-19 typically presents symptoms common in many respiratory infections, including fever and cough. However, in many cases, it progresses to a more severe disease which may include acute respiratory distress, disseminated disease, and death [1,2,3]. The SARS-CoV-2 spike protein is a homotrimeric transmembrane glycoprotein composed of S1 and S2 subunits. When the spike protein interacts with specific receptors on host cells, the receptor-binding domain (RBD), located at the C-terminus of the S1 subunit, can bind and interact with target host cell receptors, such as angiotensin-converting enzyme 2 (ACE2) [4,5,6,7]. Studies have shown that the RBD is the main target of most of the neutralizing activity in immune sera, suggesting that the RBD may be a potential target for a 2019—nCoV vaccine or therapy [7,8,9].

Currently, the available platforms for COVID-19 vaccines include inactivated vaccines, live attenuated vaccines, recombinant protein vaccines, viral vector vaccines, and nucleic acid vaccines [10]. According to Haitong Securities Research Institute’s research on global key vaccine companies and product revenue (USD 100 million) in 2021, the revenue of COVID-19 vaccine products from key global enterprises showed that Pfizer and German BioNTechnology’s mRNA vaccine revenues were USD 367.8 billion and USD 213.6 billion, respectively, accounting for more than half of the global vaccine market share. Pfizer’s mRNA vaccine accounted for the highest proportion because it rapidly attracted widespread attention and was promoted due to its safety, high efficiency, and short production cycles. Compared with other vaccine technology platforms, the mRNA vaccine platform also has advantages, including 1) A short R&D cycle; 2) Convenient large-scale production [11,12]; 3) No risk of infection or genome integration [13,14,15]; 4) The human expression system expresses antigens with good immunogenic recovery [16]; 5) The stimulation of humoral and cellular immunity [17,18]; 6) No adjuvant, etc. However, mRNA vaccines are also flawed, with high technical barriers and delivery technology patent restrictions; they are also unstable and inconvenient to preserve and transport. Traditionally, they require intramuscular injection and a professional to administer them.

Most vaccines currently undergoing clinical trials are injected intramuscularly or subcutaneously, restricting immune activation to the few draining lymph nodes [19,20]. Oral administration is thought to have a higher safety profile, better patient compliance, and lower medical costs than injection [19,21]. The complications posed by the complex gastrointestinal environment and intestinal epithelial barriers have limited the use of oral vaccines. To interact with the abundant immune cells in the lamina prima, an ideal oral vaccine must tolerate the gastrointestinal environment and overcome intestinal epithelial barriers [22].

Exosomes are cell-derived, membranous vesicles present in nearly all bodily fluids. With sizes ranging between 30 and 150 nm in diameter, exosomes are composed of a phospholipid bilayer derived from the membrane of the cell of origin. They have recently gained attention due to their natural role of shuttling molecular cargos (e.g., DNA, small RNAs, proteins, and lipids) between distant cells in the body. It has been confirmed that bovine milk is rich in exosomes which exhibit a similar potential to serve as drug-delivery nanocarriers [23]. Milk is a more affordable and accessible source compared to cell culture media. Moreover, milk-exos may provide additional benefits as naturally desirable oral delivery carriers, which indicates that milk-exos constitute a more convenient and patient-friendly therapeutic modality [24]. Taking into account the biological properties of milk exosomes and overcoming the technological challenges of mRNA vaccines, here, we described a method for the creation of milk exosomes that are loaded with RBD mRNA, evaluated their effectiveness in delivering functional mRNA, developed a novel oral vaccine technology platform based on milk exosomes, and evaluated the preliminary efficacy of the novel oral vaccine based on milk exosomes in mice to elicit humoral immunity to SARS-CoV-2 spike proteins.

## 2. Materials and Methods

### 2.1. Cell Lines and Cell Culture

293T cells (SCSP-502) (human embryonic kidney epithelial cells) were obtained from the Cell Bank of the Typical Culture Preservation Committee, Chinese Academy of Sciences (Shanghai, China). 293T cells were grown in DMEM supplemented with 10% fetal bovine serum (FBS) and 1% penicillin/strep solution at 37 °C and a 5% CO_2_ concentration in a humidified atmosphere. The pooled cells were maintained in DMEM containing 10% (vol/vol) FBS, and the medium was changed every other day. When the confluence of the cultured cells reached 80%, they were detached via treatment with 0.25% (wt/vol) trypsin and 0.1% (wt/vol) ethylenediaminetetraacetic acid (Gibco) and reseeded at a density of 1 × 10^4^ cells per cm^2^. The cultured cells before passage two were used for the experiments. For RBD mRNA expression in vitro, cells were transfected with mRNA using Lipofectamine Messenger MAX, as suggested by the manufacturer (ThermoFisher Scientific, Waltham, CA, USA).

### 2.2. Density-Gradient Ultracentrifugation Purification of Milk-Exos (DC-Milk-Exos)

Milk exosomes were isolated via density-gradient ultracentrifugation (DC). Briefly, casein-free whey was centrifuged at 100,000× *g* (Beckman Coulter, Brea, CA, USA) for 105 min to precipitate the milk-exos and resuspended in 1 mL of PBS. The concentrated milk-exos were subjected to the top of a discontinuous density gradient consisting of 2 mol/L, 1.65 mol/L, 1.3 mol/L, 0.95 mol/L, and 0.6 mol/L sucrose (7 mL volume for each angle) in 250 mmol/L Tris-HCl solution (pH 7.4). They were centrifuged at 100,000× *g* at 4 °C for 20 h. The milk-exos fraction between fractions 3 (1.3 mol/L) and 4 (1.65 mol/L) was collected. To remove sucrose, the fraction was diluted in PBS to a final volume of 40 mL and centrifuged at 100,000× *g* at 4 °C for 105 min. The pellet was resuspended in 1 mL of PBS. The DC-milk-exos were stored at −80 °C before use.

### 2.3. Characterization of the DC-Milk-Exos Morphology via Transmission Electron Microscopy

The DC-milk-exos morphology was characterized by transmission electron microscopy (TEM), as described previously [25], with some modifications. First, the DC-milk-exos (100 μg/mL) were fixed by mixing with an equal volume of 4% (*w*/*v*) paraformaldehyde at room temperature for 15 min. The selected sample (10 μL) was then subjected to a formvar-carbon-coated TEM grid and kept at room temperature for 3 min. The grid was stained by adding 10 μL of uranyl oxalate solution (4% uranyl acetate, 0.0075 mol/L oxalic acids, pH 7). The stained grid was investigated using a Hitachi 7800 transmission electron microscope operated at 120 KV and 50,000 magnification.

### 2.4. Measurement of the Particle Size Distribution of DC-Milk-Exos via NanoFCM

The particle size and number of milk-exos samples were characterized using the NanoFCM instrument (NanoFCM Inc., Xiamen, China) following the operations manual. A silica nanosphere cocktail (Cat. S16M-Exo, NanoFCM Inc., Xiamen, China) containing a mixture of 68 nm, 91 nm, 113 nm, and 155 nm standard beads was used to adjust the instrument for particle size measurement. The instrumental parameters were set as follows: Laser, 10 mW, 488 nm; SS decay, 10%; Sampling pressure, 1.0 kPa; Sampling period, 100 μs; Time to record, 1 min.

### 2.5. Proteomics

The proteomics of milk-exos were analyzed via LC–MS/MS using Easy NLC 1200-Q Exactive Orbitrap mass spectrometers (ThermoFisher). The nano-HPLC system was equipped with an Acclaim PepMap nano-trap column (C18, 100 Å, 75 μm × 2 cm) and an Acclaim Pepmap RSLC analytical column (C18, 100 Å, 75 μm × 25 cm). Typically, 1 μL of the peptide mix was loaded onto the enrichment (trap) column at an isocratic flow of 5 μL/min of 2% CH3CN containing 0.1% formic acid for 5 min before the enrichment column was switched in-line with the analytical column. The eluents used for the LC were 2% CH3CN/0.1% (*v*/*v*) formic acid (solvent A) and 100% CH3CN/0.1% (*v*/*v*) formic acid (solvent B). The gradient used was 3% B to 25% B for 23 min, 25% B to 40% B in 2 min, 40% B to 85% B in 2 min, and maintained at 85% B for 2 min before equilibration for 10 min at 3% B before the next injection. All spectra were collected in the positive mode using full-scan MS spectra scanning in the FT mode from *m*/*z* 300 to 1650 at resolutions of 70,000. A lock mass of 445.12003 *m*/*z* was used for both instruments. For MSMS on the QE, the 15 most intense peptide ions with charge states ≥2 were isolated with an isolation window of 1.6 *m*/*z* and fragmented via HCD with a normalized collision energy of 28. A dynamic exclusion of 30 s was applied.

The raw files were searched using Proteome Discover (version 2.1, Thermo Fisher, Germany) with Sequest as the search engine. The fragment and peptide mass tolerances were set at 20 mDa and 10 ppm, respectively, allowing for a maximum of 2 missed cleavage sites. The false discovery rates of proteins and peptides were 1 percent.

### 2.6. Western Blot

Total cellular and milk−exos proteins were extracted with RIPA lysis buffer, and the protein concentration was determined with a BCA protein assay kit (Thermo, A53226). Following that, the proteins were loaded onto SDS-polyacrylamide gel electrophoresis gels. After electrophoresis, the proteins were transferred onto a polyvinylidene fluoride membrane and blocked in 5% (wt/vol) bovine serum albumin (BSA) for 1 h at room temperature. Then, the membrane was incubated with anti-TSG101 (Abcam, ab125011), anti-CD9 (Abcam, ab92726), anti-RBD (Abcam, ab277628), or anti-GAPDH (Proteintech, 60,004−1) overnight at 4 °C. After washing in Tris-buffered saline with Tween (TBST), the horseradish peroxidase (HRP) secondary antibody was diluted 1:10,000 with 5% (wt/vol) BSA and incubated with the membrane for 1 h at room temperature. Excess secondary antibody was rinsed off the membrane with TBST, and a chemiluminescent signal was generated using the FluorChem E system (ProteinSimple, CA, USA) according to the manufacturer’s protocol.

### 2.7. Measurement of Zeta Potential

The Zeta potential of milk exosomes was measured thrice at 25 °C under the following settings: sensitivity of 85, a shutter value of 70, and a frame rate of 30 frames per second, while ZetaView software was used to collect and analyze the data.

### 2.8. Construction of An Oral Vaccine for SARS-CoV-2 RBD-Based DC-Milk-Derived Exosomes (RBD-DC-Milk-Exos)

mRNA designed to express the RBD proteins was obtained from a commercial provider (Novoprotein). The RBD mRNAs were purified using CIMmultus Oligo dT columns and resuspended in DNase- and RNase-free water using nuclease-free tips and tubes. Purified RBD mRNAs were prepared for loading into DC-milk-exos by preincubating them with cationic lipids. We first mixed RBD mRNA with DOTAP cationic lipids [26,27] to generate lipid-coated mRNAs. Then lipid-mRNAs were loaded into purified DC-milk-exos via mixing and incubation to construct an oral vaccine. Both processes are driven by the attractive force of water, resulting in the encapsulation of lipid-mRNAs into exosomes and exosome membranes [28]. The morphology, particle size distribution, and zeta potential of RBD mRNA-loaded DC-milk-exos were characterized via TEM, the NanoFCM instrument, and Zeta Pals, respectively. RBD mRNA loading efficiency was detected via Taqman-based quantitative real-time PCR (qRT-PCR). The RBD mRNA expression was measured using a Western blot and an enzyme-linked immunosorbent assay (ELISA).

### 2.9. RNA Extraction and Taqman-Based RT-qPCR Assay

Total RNA was extracted from the RBD-DC-milk-exos using TRIzol (Life Technologies, Carlsbad, CA, USA). First-strand cDNA was synthesized using the PrimeScript^TM^ RT Kit (TaKaRa, Beijing, China) and was used as a template to determine the expression of RBD genes with the indicated primers and probes. The following primer sequences were used: RBD, forward 5′-CTCCAGGGCAAACTGGAAAG-3′ and reverse 5′-AATTACCACCAACCTTAGAATCAAG-3′, probe, CCAGATGATTTTACAGGCTGCGTTATAG, using the Premix Ex Taq™ (Probe qPCR) reagent (TaKaRa, Beijing, China) in qRT-PCR. The cycling parameters were as follows: a PCR reaction was carried out on 50 ng cDNA samples using 0.2 μmol/L of each primer, 0.4 μmol/L RBD probe, and 10 μL 2×Premix Ex Taq Mix. The following conditions were used: 95 °C for 30 s, 40 cycles at 95 °C for 5 s, and 60 °C for 31 s in a Thermo Fisher QuantStudio 5 and analyzed with the dedicated software.

### 2.10. ELISA Assay

The cell lysate, culture supernatant, and serum were homogenized for protein extraction. The spike protein RBD (Beyotime, Shanghai, China) and neutralizing antibody (Vazyme, Nanjing, China) expressions were determined using ELISA kits according to the manufacturer’s instructions. The antibodies detected using the Vanzyme ELISA kits were neutralizing or would inhibit the binding of RBD to its receptors, which indicates that it is a competitive binding ELISA.

### 2.11. Measurement of RBD Activity In Vitro

We tested whether an oral vaccine for SARS-CoV-2 RBD-based DC-milk-derived exosomes (RBD-DC-milk-exos) could deliver functional RBD mRNA into 293T cells. For in vitro studies, the oral vaccine was added to the resulting formulations of cultures of human cells, and the cells were grown overnight to allow for RBD mRNA uptake and expression. Then, the cells were assayed for RBD activity via Western blot and ELISA.

### 2.12. Delivery Function Verification of RBD-DC-Milk-Exos In Vivo

We used age-matched female BALB/c mice (Speford Biotechnology Co., Ltd., Beijing, China). All animal experimentation was performed following institutional guidelines for animal care and use and was approved by the Experimental Animal Ethics Committee of Youji (Tianjin, China) Pharmaceutical Technology Co., Ltd. (IACUC-20220726-05.00). Female BALB/c mice (23~26 g, Speford, Beijing, China) were housed in a pathogen-free facility with standard conditions of a temperature of 24 °C, a 12-h light/dark cycle, and food and water ad libitum. Before establishing the mice model of vaccines, the mice were randomly divided into a control group (N = 2) and an exosome-delivered RBD mRNA group (N = 2) in the preliminary experiment. All mice were sacrificed after two days with 1% isoflurane. Duodenum was collected on day 2. The expression of RBD was detected via ELISA analysis in the duodenum. The mice were employed to study the activity of RBD mRNA-DC-milk-exos administered via duodenal injection. The mice were randomly divided into 2 groups: 1) the control group (saline, 1000 μL) (N = 3); and 2) the RBD-DC-milk-exos (0.5 mg mRNA/1000 μL) group (N = 5). All treatments began with duodenal injections on days 1, 15, and 36. All mice were sacrificed two weeks after the last treatment with 1% isoflurane. Blood was collected at different time points (days 0, 7, 14, 21, 28, 35, 42, and 49). Verifying neutralizing antibodies of the oral vaccine for SARS-CoV-2—based milk-derived exosomes in vivo was executed via ELISA analysis in serum. N = 3 (control), N = 5 (RBD mRNA-DC-milk-exos), and the data are presented as mean ± STD. ** *p* < 0.01, *** *p* < 0.001 vs. control group.

### 2.13. Statistical Analyses

All data are presented as mean ± S.D. via Graph Pad prism 5 and were analyzed by unpaired Student’s *t*-tests with SPSS 19 statistical software (SPSS Inc., Chicago, IL, USA). *p* < 0.05 was considered statistically significant for group differences. An asterisk (*) represents *p* < 0.05; a double asterisk (**) represents *p* < 0.01; a triple asterisk (***) represents *p* < 0.001.

## 3. Results

### 3.1. Preparation of Bovine-Milk-Derived Exosomes

To evaluate our preparation methods, bovine-milk-derived exosomes (milk-exos) were isolated and purified via density gradient ultracentrifugation (DC) (Figure 1A). Six components were collected via DC, among which F1 was about 5 mL, and F2 to F6 were all 7 mL (Figure 1B). To characterize the isolated milk-exos morphology, biomarkers (CD9 and TGS101) were used to determine the exosome-containing fraction. Exosomes were mainly concentrated in F3 and F4 (Figure 1 C,D). The morphology exhibited a rich profusion of mixed populations of exosomes with predominantly intact vesicles consistent with classical exosome-like morphology and a typical cup-like structure (Figure 1D). Milk exosomes are characteristic of exosomes in the range of 30–150 nm in diameter and were observed in 0.95 mol/L–1.30 mol/L sucrose. As fractions 3 and 4 were enriched in exosomes, they were pooled together for further analysis.

### 3.2. Purification and Characterization of Bovine-Milk-Derived Exosomes via Density Gradient Ultracentrifugation

The morphologies of pooled exosome fractions exhibited a rich profusion of exosomes consistent with a classical exosome-like morphology (Figure 2A), size distribution (Figure 2B), and protein markers (Figure 2C). A three-way Venn diagram of proteins revealed 1022 proteins common to all datasets, and 961 proteins were commonly identified in all three DC-milk-exos, as shown in Figure 2C. Proteomics analysis of the samples showed that exosome protein lysates were prepared and verified with cluster analysis for vital exosomal membrane markers CD9, CD63, CD81, and TSG101. The absence of the microvesicle surface markers, GM130, and calnexin, as well as the absence of the endoplasmic reticulum (ER) marker, calnexin, confirmed that the isolated milk-exos were not contaminated with other multivesicular bodies (Figure 2C). Three batches of bovine milk-derived exosomes obtained via density gradient ultracentrifugation (DC-milk-exos) were analyzed, and the results indicated that there were no differences among multiple batches of DC-milk-exos.

### 3.3. Loading of RBD mRNA Milk-Derived Exosomes

To determine whether DC-milk-exos could be loaded with exogenous, in vitro synthesized mRNAs, we designed and synthesized a test receptor-binding domain (RBD) mRNA encoding immunogenic forms of the SARS-CoV-2 spike. The RBD coding sequence region (CDS) is 675 base pairs (bp) long and has the FLAG tag. Examining the in vitro synthesized RBD mRNA using a bioanalyzer (Agilent) confirmed that the RBD mRNA sample ran as a single band of 1100 bps (Figure 3A), consistent with the size that we expected to design in vitro transcription. To assess its functionality, the RBD mRNA was transfected into 293T cells, and RBD peptide expression was investigated the next day via Western blot (Figure 3B). These results indicated that RBD mRNA was synthesized according to in vitro transcription (IVT) and had a translational function that could be translated into RBD protein in cells.

To load the IVT RBD mRNA into DC milk-exos, we first mixed it with cationic lipids (DOTAP) to generate lipid mRNAs. Then, we loaded the lipid-mRNA into DC-milk-exos via mixing-induced partitioning (Figure 4A). Both processes are driven by the attractive force of a charge, which resulted in the encapsulation of lipid-mRNAs into milk-exos membranes here. To biophysically characterize the RBD mRNA milk-exos, morphology, size distribution, and zeta potential analyses were performed (Figure 4B–D). To determine the loading efficiency of this process, the products of three independent mRNA-loading reactions were examined. A TaqMan—based real-time quantitative PCR assay showed that the loading efficiency reached 57.3% (Figure 4E).

### 3.4. Preliminary Assessment of Oral Vaccines for RBD mRNA-DC-Milk-Exos In Vitro and In Vivo

Next, we tested whether milk-exos loaded with RBD mRNA could deliver functional RBD mRNA into human cells. The Western blot and ELISA results established that RBD mRNA-loaded milk-exos could deliver mRNA into 293 cells and produce RBD peptide 24 h later (Figure 5 A,B, *p* < 0.01).

To further confirm the ability to stimulate neutralizing antibodies, RBD mRNA-milk-exos were injected into the duodenum (i. d.) of 9–11-week-old female BALB/c mice (Figure 6A). Blood (0.1 mL) was collected on days 0, 7, 14, 21, 28, 35, 42, and 49 for antibody detection. Using ELISA kits adapted for detecting mouse-derived antibodies, we observed that vaccinated animals produced a relatively constant level of neutralizing antibodies against RBD after the second injection (Figure 6B). The expression of RBD was detected via ELISA analysis in the duodenum. The results show that RBD protein was expressed in the duodenum in the preliminary experiment (data not shown).

## 4. Discussion

In this study, we evaluated whether an oral vaccine for SARS-CoV-2—based bovine-milk-derived exosomes could deliver functional mRNA in vitro and in vivo and induce anti-S antibody responses. We first successfully verified the technical feasibility of oral mRNA delivery based on bovine-milk-derived exosomes as delivery vehicles.

Compared with injection, oral administration is generally considered to have a better safety profile, better patient compliance, and lower medical costs [19,21]. Like liposomes, exosomes have a bi-lipid membrane and an aqueous core; therefore, they can be loaded with hydrophilic and lipophilic drugs [29]. However, the practical application of exosome-based therapeutics in clinical transformation remains an ongoing challenge. Many studies have isolated exosomes from cell culture media with a low yield and cost, making scaling-up production difficult [30]. Although density-gradient ultracentrifugation has been the gold standard for exosome isolation and purification, several studies have been carried out regarding the extraction of exosomes from cow’s milk using differential centrifugation alone [31,32] or precipitation [33,34] techniques. In conjunction with differential centrifugation, we employed density-gradient ultracentrifugation methods to isolate exosomes from milk that were essentially free of contamination from microvesicles. The final sample purity could reach 100%. However, the technique used in this study to purify exosomes is incompatible with large-scale exosome production, which is one of the obstacles to the progress of industrialization. The good news is that we established a chromatography-based novel manufacturing process to purify exosomes from bovine milk with better quality than those produced via DC (data are not shown). It was reported that bovine-milk-derived exosomes exhibited a similar potential to serve as drug-delivery nanocarriers [23]. Milk is a more affordable and accessible source compared to cell culture media. Moreover, bovine-milk-derived exosomes might provide additional benefits as naturally desirable oral delivery carriers, which indicates that bovine-milk-derived exosomes constitute a more convenient and patient-friendly therapeutic modality [24].

Herein, we presented a preliminary assessment of an oral vaccine that differed from previous nanomaterials-based oral vaccine delivery technologies in that it comprised bovine-milk-derived exosomes. We showed the capabilities of milk-derived, exosome-derived oral COVID-19 mRNA vaccine in delivery technology and industrialization. Han et al. showed that ileal administration could be used to evaluate intestinal mucosal absorption while excluding the notable pH and enzyme degradation factors found in the stomach and duodenum in the delivery of oral insulin [35]. As a reference, the enteric delivery of mRNA vaccine candidates could be another strategy in addition to the oral route. Although the results from this study suggested the beneficial effects of an oral vaccine for the prevention of the novel coronavirus and offered a potential regimen for clinical application, we need to expand the number of animals used in the trial and further verify the efficacy of the oral vaccine in different animal models (such as large animals) in future studies. Furthermore, its precise molecular mechanisms and safety need to be investigated further. Currently approved SARS-CoV-2 mRNA-LNP—based and vector—based vaccines rely on intramuscular administration, which induces high levels of circulating antibodies, memory B cells, and circulating effector CD4+ and CD8+ T cells in animal models and humans. However, parenteral vaccines do not induce high levels of potent antiviral immune memory at sites of infection, such as tissue-resident memory T cells (TRM) and B cells (BRM), as well as mucosal IgG and dimeric IgA. This contrasts the SARS-CoV-2 infection in humans and mice, in which CD8+ TRM are robustly induced [36]. Therefore, their effectiveness is still suboptimal, and a new paradigm of oral vaccine technology is urgently needed.

In the present study, the working environment of the oral vaccine was the intestine, which is more biocompatible as an application strategy. Oral vaccines are ideal for capsule or tablet administration. Because a liquid preparation vaccine was used in this study, we intend to first perform the route of administration into the duodenum to verify the efficacy of milk-derived exosome-mediated mRNA in vivo. At the same time, we are now considering the development of an oral vaccine filled with solid preparations into capsules.

## 5. Conclusions

SARS-CoV-2 RBD mRNA vaccine loading via bovine-milk-derived exosomes is an easy, cheap, and novel way to introduce immunity against SARS-CoV-2 in vivo. In the near future, an mRNA delivery system based on milk-derived exosomes will serve as a platform for mRNA therapeutics development.

## Figures and Tables

**Figure 1 vaccines-11-00673-f001:**
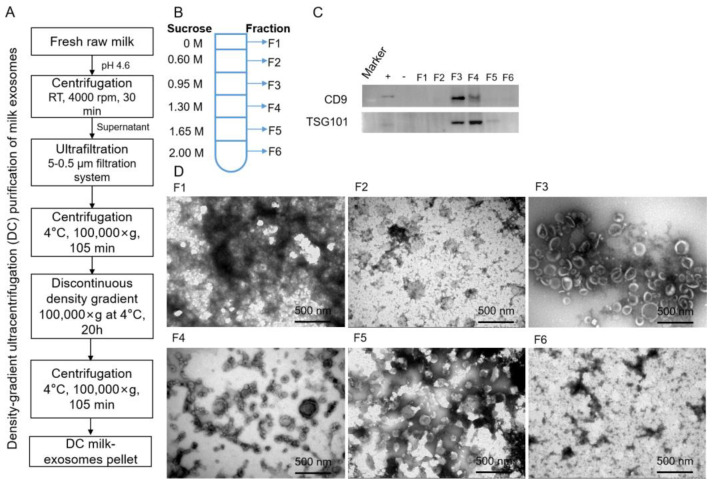
Identification of bovine-milk-derived exosomes in fractions of density gradient ultracentrifugation. (**A**) Schematic representation of the major steps involved in isolating exosomes from bovine raw milk. (**B**) F1–F6 represents the corresponding concentrations of sucrose. (**C**) The exosome suspension was analyzed via Western blot. Immunoblots showed exosome markers in different milk exosome fractions. +, positive control (the protein of HaCat cells); −, negative control (the protein of Hela cells). (**D**) The morphologies of different fractions obtained via TEM. Scale bars = 500 nm.

**Figure 2 vaccines-11-00673-f002:**
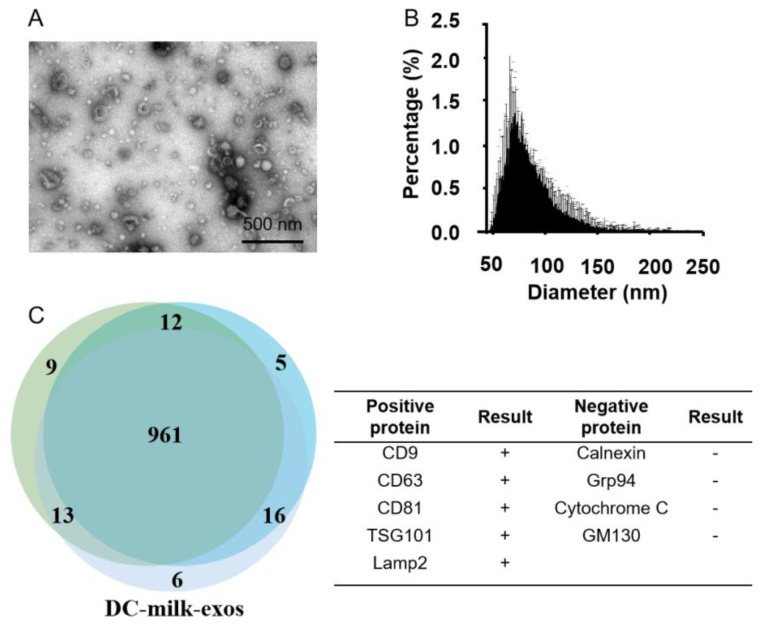
Characterization of DC-milk-exos. (**A**) The morphology and (**B**) the particle size analysis were detected via TEM and nanoFCM, respectively. (**C**) A three-way Venn diagram of proteins from three batches of DC-milk-exos revealed 1022 proteins common to all datasets. Cluster analyses for vital exosomal membrane markers CD9, CD63, CD81, and TSG101, microvesicle surface markers GM130, and endoplasmic reticulum (ER) marker calnexin are indicated in the table. Abbreviations: TEM, transmission electron microscope; DC-milk-exos, bovine-milk-derived exosomes via density gradient ultracentrifugation. Scale bars = 500 nm.

**Figure 3 vaccines-11-00673-f003:**
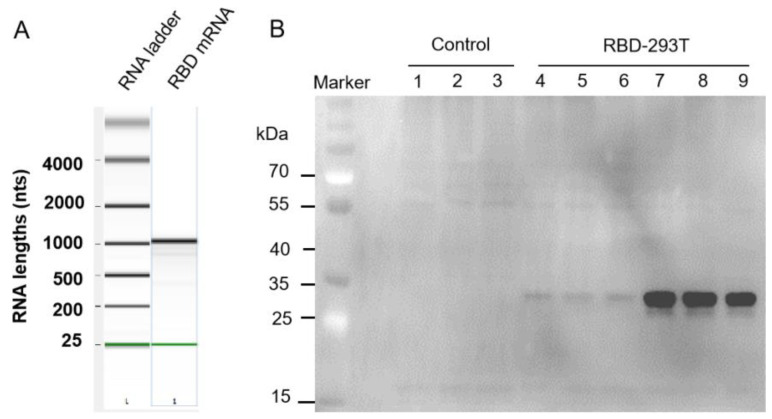
Characterization of SARS-CoV-2 receptor-binding domain (RBD) mRNA. (**A**) Gel-like image of in vitro synthesized RBD mRNA interrogated using an RNA chip on an Agilent Bioanalyzer. Data for RNA markers, RBD mRNA, are presented from left to right. (**B**) Western blot for SARS-CoV-2 RBD protein expression in 293T cells when the RBD mRNA was transfected into 293T cells with an additional 24 h of treatment along with 1 μg and 3 μg of RBD mRNA. Lanes 1–3: control; Lanes 4–6: 1 μg of RBD-293T; Lanes 7–9: 3 μg of RBD-293T.

**Figure 4 vaccines-11-00673-f004:**
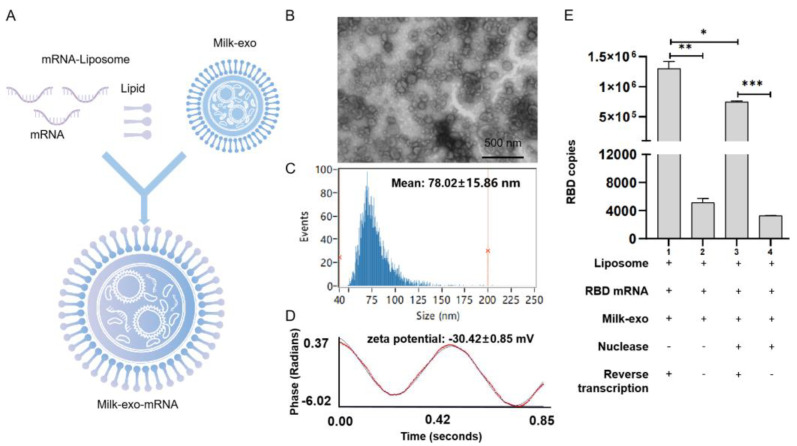
Characterization of milk-derived exosome—based vaccine for SARS-CoV-2. (**A**) Flowchart of vaccine preparation for SARS-CoV-2. The morphology (**B**), particle size distribution (**C**), zeta potential (**D**), and RBD mRNA loading efficiency (**E**) were determined. Scale bars = 500 nm.

**Figure 5 vaccines-11-00673-f005:**
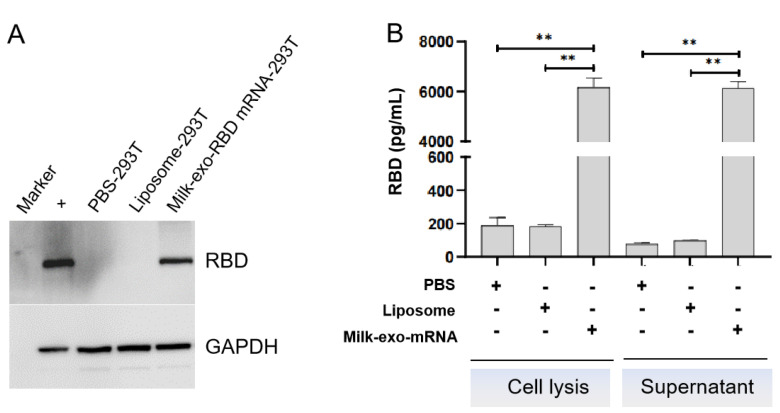
mRNA-loaded exosomes deliver functional SARS-CoV-2 RBD mRNA to human cells in vitro. (**A**) Western blot analysis of the expression of RBD mRNA delivered by oral vaccine in 293T cells. (**B**) The ELISA detection of RBD expressed in 293T cell lysate and secreted into the culture supernatant 24 h after the milk-derived exosome-based vaccine transfection. The data are presented as mean standard deviation with a group size of three. ** *p* < 0.01 vs. PBS.

**Figure 6 vaccines-11-00673-f006:**
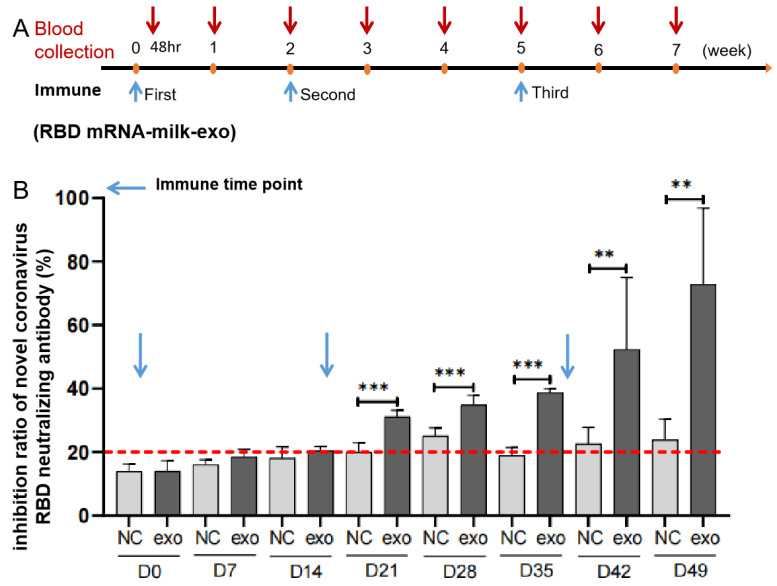
Validation of SARS-CoV-2 RBD mRNA-loaded milk exosomes delivering functional SARS-CoV-2 RBD mRNA in vivo. (**A**) Mouse immunization and sera sampling schedule. BALB/c mice received the same doses of an oral vaccine for the SARS-CoV-2—based milk-derived exosomes (N = 5) or the control saline (N = 3) on day 0 and were boosted again on days 14 and 35. Sera were collected on days 0 (pre-vaccination), 7, 14, 21, 28, 35, 42, and 49 (post-vaccination). The blue and red arrows represent the time points of immunization and blood collection, respectively. (**B**) Preliminary assessments for neutralizing antibodies of the oral vaccine for SARS-CoV-2—based milk-derived exosomes in serum were determined via ELISA. The red dotted line represents the cutoff value of neutralizing antibodies against the RBD peptide. N = 3 (control), and N = 5 (RBD-DC-milk-exos), and the data are presented as mean ± STD. ** *p* < 0.01, *** *p* < 0.001 vs. control group.

## Data Availability

All data are available in the main text. Further inquiries can be directed to the corresponding author.

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
