# Peer review of "Intraduodenal Delivery of Exosome-Loaded SARS-CoV-2 RBD mRNA Induces a Neutralizing Antibody Response in Mice"

_vaccines, 2023, doi:10.3390/vaccines11030673_

Round 1

Reviewer 1 Report

Title: An oral vaccine for SARS-CoV-2 RBD mRNA-bovine milk-derived exosomes induces a neutralizing antibody response in vivo.

General Comments: The authors present a report that details the isolation of highly pure exosomes from bovine milk using density gradient ultracentrifugation. The isolated exosomes were successfully loaded with mRNA coding for the receptor binding domain of SARS-CoV2. In a mouse model, intraduodenal administration of mRNA-loaded exosomes using a single-prime-double-boost protocol resulted in the induction of detectable neutralizing antibody 7 days after the first booster dose. The authors propose exosome-loaded mRNA as a viable approach for vaccination against this viral infection. This is a very interesting study that introduces the possibility of using cow milk derived exosomes as a vehicle to deliver mRNA vaccines. The findings should be interesting to the readership of Vaccines.

Specific Comments:

1.       The procedure for generating exosomes from bovine milk is well described. However, Figure 5b showing the expression of SARS-CoV2 RBD in cell pellets and supernatants could be confusing without appropriate labeling (or explanation) of the y-axis. Are these densitometric measurements of Western blot images? The authors need to make this clear.

2.       The rationale for choosing an oral route for delivering mRNA-loaded exosomes is unclear. Although, like several other coronaviruses, fecal shedding of SARS-CoV2 has been extensively reported. However, the protective effect of primary infection via the fecal-oral route on protection from respiratory infection has not been fully elucidated. The most intuitive approach to delivering a vaccine meant to induce mucosal immunity would have been through the intranasal route. The authors did not justify why they would prefer the enteric route to intranasal route as a practical method to deliver a SARS-CoV2 mRNA vaccine.  This problem is underscored by the results of other workers who demonstrated that intranasal delivery of exosome-loaded SARS-CoV2 mRNA generated a robust antibody response (Wang et al., 2022, PMID:35788687; Popowski et al, 2022, PMID: 35847197). Why did the authors choose intraduodenal, rather than intranasal, delivery of this exosome-based vaccine candidate?

3.       Intraduodenal delivery of this vaccine candidate does not seem to generate significant immune response. Antibody was only detectable at a significant level only 7 days after receiving a booster. It also does not appear that the memory response is robust.

4.       The strength of this manuscript is the idea of bovine milk as a source of exosomes and the method used to successfully isolate them. However, the choice of enteric delivery of the vaccine candidate is a major flaw that almost totally neutralizes the impact of the work, especially in light of other studies on exosome-based SARS-Cov2 vaccines. At the minimum, the authors need to show that RBD is expressed in the epithelial lining of the gut or in the draining lymph nodes of the gut. It is also important to know what types of antibodies were generated (IgA vs IgG).

Author Response

Dear Editor,

Thank you for giving us the opportunity to resubmit our manuscript vaccines-2183121 entitled “An oral vaccine for SARS-CoV-2 RBD mRNA-bovine milk-derived exosomes induces a neutralizing antibody response in vivo”. We have revised the paper in response to reviewers’ comments. The changes are marked using the “Track Changes” function in the text of the revised version. The main corrections in the paper. A point-by-point response to the reviewer’s comments, please see the attachment.

Reviewer 2 Report

The study done by Zhang and Wang et al. is of interest in a growing field of mRNA research. The idea of finding a orally bioavailable vaccine, particularly against respiratory viruses like SARS-CoV-2, is of high importance, especially considering the current licensed vaccines do not protect from infection.

It is clear to this reviewer that the authors have done a lot of work to isolate and characterize these milk exosomes and that deserves praise. The overall story is nice and the application of this technology can be useful in the vaccine and immunological field. Overall, the study is decently written, albeit with some English language and grammatical issues. The figures are nice and relatively easy to follow but perhaps the figure legends could be hard for lay readers. While nice, the paper is not without issues:

Major issues:

There is a huge flaw in this study because the authors cannot say "oral vaccine" yet. The in vivo data is from a duodenal injection and bypasses the oral cavity and stomach. Thus, there is a lack of evidence that this vaccine can be taking orally. The only evidence presented is that it can be taken in by the GI tract. To be a truly "oral" vaccine, the authors need to assess intake through the mouth. To amend this, I would suggest that the authors change the study to a preliminary assessment of an 'oral' vaccine and write about it as such. 

Overall, the discussion is lacking. I would use the paragraph from lines 348-358 to follow on from the limitations addressed from using exosomes (lines 330-332). The authors perform a lot of work to show that this application works to induce antibodies and I think the discussion can build from this. What type of antibodies? are there T cells as well (easy to look for since it's only RBD epitopes)? Is this immunity better than already licensed vaccines? Does this immunity protect better from viral challenge? All these questions are important to address if this application is be developed.

Similarly, the conclusion is lacking. The conclusion should be that this is an easy, cheap and novel way to introduce immunity against SARS-CoV-2 (i.e. has advantages over current technology). In no way have the authors shown that this can prevent SARS-CoV-2 infection and I would remove this from the conclusion. 

On a statistical note, why did the authors use a mean and SD? Is the data normally distributed? If yes, it is not indicated how the authors checked this. Generally, smaller sample sizes like this should be represented by a median.  

Minor issues:

On line 30, it should be "When the spike protein..."

The sentence on lines 30-33 should be rewritten. The English is slightly confusing. Something like "The receptor binding domain (RBD), located at the C-terminus of the S1 subunit, can bind and interact with target host cell receptors, such as angiotensin-converting enzyme 2 (ACE2)." may be better for readers. 

On lines 33 and 34, it should be "the RBD". 

On lines 33-35, "Studies have shown that RBD is the main target of the most neutralizing activity in immune serum, suggesting that RBD may be a potential target for the 2019-nCoV vaccine or therapy". - The references provided here do not provide any evidence of what is said in this sentence. Two of them are from pre-pandemic era (2003 and 2005) on the original SARS. There is a plethora of literature out that is much more relevant to show evidence of what is said in this sentence. I would suggest that the authors use more relevant and up-to-date references. 

On lines 38-41, "In 2021, the revenue of COVID-19 vaccine products..." 
This sentence needs a reference/source of this information. 

On lines 44-45, "an excellent immune effect" - what is meant by this? This is a bit of a nebulous term. Immune effect on what? If the authors mean induction of immune responses, they cover that on lines 47-48 with "stimulation of humoral and cellular immunity". 

On lines 52-58, I would also argue that an ideal oral vaccine, particularly against respiratory viruses, should be able to be delivered into the respiratory system (like flu mist). As a suggestion, this may be a beneficial point to make as intramuscular vaccines do not induce mucosal-level immunity. 

On line 90, it should be "the casein free whey" (no the).

On line 101, a reference is missing with "as described before". 

In section 2.8, it is not clear how the lipid-mRNA was put into the milk-exos. I'm guessing that adding the lipids to the mRNA allows it to fuse into the exosomes as mentioned by "Tsai, S.J. et al. Journal Biological Chemistry, 2021". It is also important to mention that some of the material written from section 2.8 matches exactly what is written in the aforementioned publication. To make it clearer, I would suggest adding something small that indicates that the lipids are able to deliver the mRNA into the exosomes. It is shown in Figure 4A very nicely, so perhaps a sentence that summarizes this would make it clearer. 

In section 2.11., the in vitro studies were carried out in "human cells". Given that the authors have only mentioned 293T cells, I am guessing it was done in these cells. For reader clarity I would instead write "into 293T cells" on line 191. 

On lines 222-224, "To characterize the isolate milk-exos biophysically..." this sentence doesn't make complete sense. I would suggest that the sentence be rearranged to something clearer such as, "To characterize the isolated milk-exos morphology, biomarkers (CD9 and TGS101) were used to determine the exosome containing fraction". - please note that there is a typo in the parenthesis here, as in the text it says TGS 110 and I believe it is meant to be TGS 101.

On line 240, should it be "pooled exosomes fractions"?

On line 295, should it be "293cells"? 293 cells are different to 293T cells. 

Figure 5B has no Y-axis. This should not be left to the reader to interpret. It is not clear what these numbers are.

For Figure 6B, it is not clear what the "inhibition ratio" is on the Y axis. Again, this should not be left to the reader to interpret. While this is shown in the ELISA kit instructions, the authors need to make it clearer to the current readers. It's also hard to gauge how effective the induction of these nAbs are because there is no positive control group (such as Pfzier BioNTech mRNA). 

On lines 321-323, the authors mention that they demonstrated that an "oral vaccine" could induce anti-S responses. The authors need to be more specific. They demonstrated a preliminary assessment of an 'oral' vaccine that could induce anti-RBD antibodies in vitro and in vivo. I would be careful with the term “neutralizing” here. This is because, although the ELISA performed shows RBD-ACE2 blocking (thus assuming neutralization), there is no evidence of blocking of whole or pseudotyped viral particles.

On line 325, "administration was" should be "administration is". Unless of course it was previously considered and that thought has since changed. 

On line 327, "could potentially be" should be replaced with "can be loaded" as the authors have shown evidence for this in this paper, as well as others.

On line 329, "remained" should be "remains". 

On line 330, "Many studies isolated" should be "Many studies have isolated". 

On lines 333-338, this should be integrated with the paragraph above to show the good side of using exosomes in more recent years of research. 

On line 344, "needed" should be "need". 

Following on the end of the sentence in line 347, the authors need to discuss the application of applying the vaccine orally (i.e. through the mouth) and not injection into the duodenum if they want to call it an "oral vaccine". 

Author Response

Dear Editor,

Thank you for giving us the opportunity to resubmit our manuscript vaccines-2183121 entitled “An oral vaccine for SARS-CoV-2 RBD mRNA-bovine milk-derived exosomes induces a neutralizing antibody response in vivo”. We have revised the paper in response to reviewers’ comments. The changes are marked using the “Track Changes” function in the text of the revised version. A point-by-point response to the reviewer’s comments, please see the attachment.

Round 2

Reviewer 1 Report

In response to the original review, the authors have done the following:

1. Provided appropriate labels to Figure 5B and explained that this was not a densitometric measurement but ELISA detection of RBD expressed in 293T cell lysate or secreted into the  culture supernatant.

From this response, the authors have successfully demonstrated that the mRNA they packaged into the exosome system was functional and could lead to the expression of the RBD in an in vitro system.

2. The authors attempted to provide a justification for an oral vaccine - cheap and easy to administer. This is a commendable effort. However, the concern about this study is not about the justification for an orally-deliverable exosome-based vaccine but mainly about the immunological basis of the claims made as the authors attempted to interpret their data. The only evidence of possible vaccine efficacy offered by the authors is a modest increase in "neutralizing antibodies", detected using ELISA, following vaccine delivery. This conclusion caused confusion in the original manuscript because the authors did not do the following in the methods section: 

There is no clear indication that the antibodies detected using the  Vanzyme ELISA kits were neutralizing or would inhibit the binding of RBD to its receptors (although the ELISA data is presented as inhibition ratio of novel coronavirus, which indicates that they are looking at a competitive binding ELISA). The authors need to state this clearly in the methods section (Vanzyme ELISA kits is not known by everyone!).  

3. Other than the measured antibody response, the authors showed no evidence that this protein was expressed anywhere in vivo (epithelial linings of the duodenum?) There is no evidence that the protein could be found in macrophages.

This is a major weakness of this study since there is no direct link between exosome-delivered mRNA and immune response to RBD other than the presence of antibodies. 

4. There is no evidence that other arms of the immune system (T cells?) were activated by this exosome-delivered mRNA. The only evidence of antigenicity is the detection of antibodies using ELISA. Could these just be cross-reactive antibodies? The antibody class detected is unknown (could the assay be measuring cross-reactive IgM rather than RBD-specific IgG?). What is the negative control in this study? Could the authors have used scrambled mRNA containing the same ribonucleosides as RBD as control mRNA?   In addition, a simple lymphocyte proliferation assay using RBD peptides loaded into dendritic cells would have been helpful in further ensuring the claim of an immune response in this system. This is another major weakness of this study. 

5. Title: I would not use the word "vaccine" for this manuscript in its present form since that word attracts a lot of expectations that the data presented may not immediately support. The authors should consider using something like

"Intraduodenal delivery of exosome-loaded SARS-CoV-2 RBD mRNA induces a neutralizing antibody response in mice"

Author Response

Dear Review,

Thank you for giving us the opportunity to resubmit our manuscript vaccines-2183121 entitled “An oral vaccine for SARS-CoV-2 RBD mRNA-bovine milk-derived exosomes induces a neutralizing antibody response in vivo”. We have revised the paper in response to reviewers’ comments. The changes are marked using the “Track Changes” function in the text of the revised version. The main corrections in the paper. A point-by-point response to the reviewer’s comments, please see the attachment.
